# Orexins/Hypocretins and Cancer: A Neuropeptide as Emerging Target

**DOI:** 10.3390/molecules26164849

**Published:** 2021-08-11

**Authors:** Couvineau Alain, Nicole Pascal, Gratio Valérie, Voisin Thierry

**Affiliations:** INSERM UMR1149/Inflammation Research Center (CRI), Team “From Inflammation to Cancer in Digestive Diseases” Labeled by “la Ligue Nationale Contre le Cancer”, University of Paris, DHU UNITY, 75018 Paris, France; pascal.nicole@inserm.fr (N.P.); valerie.gratio@inserm.fr (G.V.); thierry.voisin@inserm.fr (V.T.)

**Keywords:** orexins, neuropeptide, GPCR, apoptosis, cancer, gastroenterology

## Abstract

Over 20 years ago, orexin neuropeptides (Orexin-A/hypocretin-1 and Orexin-B/hypocretins-2) produced from the same precursor in hypothalamus were identified. These two neurotransmitters and their receptors (OX1R and OX1R), present in the central and peripheral nervous system, play a major role in wakefulness but also in drug addiction, food consumption, homeostasis, hormone secretion, reproductive function, lipolysis and blood pressure regulation. With respect to these biological functions, orexins were involved in various pathologies encompassing narcolepsy, neurodegenerative diseases, chronic inflammations, metabolic syndrome and cancers. The expression of OX1R in various cancers including colon, pancreas and prostate cancers associated with its ability to induce a proapoptotic activity in tumor cells, suggested that the orexins/OX1R system could have a promising therapeutic role. The present review summarizes the relationship between cancers and orexins/OX1R system as an emerging target.

## 1. Introduction

The identification of orexin, also termed hypocretin, is still young in science history [1]. It was in the late 1990s, when two independent groups identified and characterized these neuropeptides in mouse hypothalamus [2,3]. Orexins (for more clarity in this text, the single term “orexin” will be retained) were encoded by the same gene (*hcrt*) constituted of two exons producing a unique precursor named prepro-orexin which provided two peptide isoforms, orexin-A (OxA also termed hypocretin-1) and orexin-B (OxB also termed hypocretin-2) [4]. The major role of orexins in the central nervous system (CNS) is to regulate wakefulness [5]. The dysregulation of orexin production in hypothalamus as well as the loss of orexins neurons leads to narcolepsy associated with cataplexy designated as narcolepsy type I [5]. [In addition to sleep regulation, orexins control energy homeostasis, reward seeking, food consumption, drug addiction and motivation [6,7]. Although the central action of orexins has been widely described, some studies have demonstrated that orexins also play a physiological role in the peripheral nervous system (6). Orexins are able to regulate reproductive and neuroendocrine functions, gastrointestinal motility, blood pressure, metabolism and energy balance. However, few reports have been dedicated to orexins’ role in the peripheral nervous system, indicating that these peripheral actions remain relatively controversial [6]. Orexins mediate these biological actions by activating two orexin receptor subtypes that have been identified as orexin receptor type 1 (OX1R) and orexin receptor type 2 (OX2R). These two receptors are associated with Gq protein and belong to the G protein-coupled receptor (GPCR) family [8]. The activation of these receptors by orexins leads to the production of intracellular Ca^2+^, involving the activation of αq, the phospholipase C and inositol triphosphate (IP3) production (Figure 1) [8]. 

Since the 2000s, it has been shown, in pathological conditions, that OX1R (but not OX2R) is abnormally expressed in peripheral cancers [9]. This ectopic expression of OX1R has been observed in inflammation states, including intestinal bowel diseases (IBD), multiple sclerosis, pancreatitis and also in digestive cancers such as colon, pancreas and liver cancers [4,10,11] and non-digestive cancers such as prostate cancer [12]. The activation of this ectopically expressed OX1R induces anti-inflammatory and anti-tumoral effects, demonstrating its putative therapeutic interest in treatment of these pathologies, in particular, in cancer [6,13]. A recent report focused on the connection between narcolepsy, Alzheimer’s and Parkinson’s diseases—where a dysregulation of orexin signaling was observed—and cancer, suggesting that the modulation of orexin signaling could have a putative therapeutic role in cancer treatment [14]. In this respect, the present review reports the anti-tumoral role of the orexins/OX1R system in various cancers.

## 2. Orexins and Orexin Receptors

Orexins (OxA and OxB) are encoded by the same mRNA which produced a common precursor [2]. OxA is a peptide of 33 amino acids that has a pyroglutamyl residue in the N-terminal position, two disulfide bridges between C6–C12 and C7–C14, and an amidated C-terminal end (Figure 1) [15]. OxB is a peptide of 28 amino acids that also has a C-terminal end amidated (Figure 1) [3]. These two peptides are highly conserved in all mammalians [15]. It should be noted that orexin-like genes are not found in invertebrates [15]. Solution structures of OxA and OxB reveal the presence of two α-helices between residues S13-G22 and G24-T32 for OxA and residues G6-S18 and H21-T27 for OxB (Figure 1). These two α-helice domains are linked by a small flexible domain (Figure 1). Structure–function relationship analyses of OxB revealed that the C-terminal domain N20-M28 is crucial for the peptide activity [16]. However, the N-terminal moiety of OxB from residue R1 to G6 is not essential for its activity [16]. As mentioned in the introduction, these two peptides are able to interact with two receptor subtypes, OX1R and OX2R. OX1R had a better affinity for OxA than OxB, whereas OX2R had the same affinity for two peptides [8]. 

Associated with the strong impact of orexins on sleep regulation, inhibition of orexin receptors represents an important therapeutic option to treat insomnia [17]. In this context, pharmaceutical industries, but also academic laboratories, have developed various antagonists that are able to regulate wake–sleep cycles [18]. Many molecules have been produced which were divided into two subclasses: the single orexin-receptor antagonists (SORAs) including selective OX1R antagonist (SORA1s) and selective OX2R antagonist (SORA2s), and the dual orexin-receptor antagonists (DORAs). Among these various compounds, three antagonists termed suvorexant [19], lemborexant [20] and lately daridorexant [21] were approved by the U.S. Food and Drug Administration (FDA) in insomnia treatment. In 2015, the first X-ray structure of OX2R complexed with one orexin antagonist (suvorexant) was determined (Figure 1) [22]. One year after, the structure of OX1R associated with suvorexant has been solved (Figure 1) [23]. As expected, orexin receptors have a similar structure to the other GPCRs [24], consisting of seven-transmembrane (TM) domains (Figure 1). It should be noted that the analysis of the structure of orexins receptors complexed with DORA shows that the backbone root-mean-square deviation (r.m.s.d.) between OX2R and β_2_AR was close to 2A, indicating a good structural similarity, although a low sequence homology was observed between these two receptors [22]. Analysis of orexin receptor structures demonstrated that the suvorexant-binding pocket was accessible to the extracellular environment [22,23]. Structure–function relationship studies associated with the 3D modeling structure of OX1R bound to OxB and molecular dynamic simulation have demonstrated that the mutation into alanine residue of K120, P123, Y124, N318, F340, T341, H344 and W345 located in the TM2, TM3, TM6 and TM7 reduced the binding affinity of OxB to OX1R and/or the ability to activate the Ca^2+^ signaling pathway [16]. Moreover, L11 and L15 residues belonging to OxB sequence could interact with OX1R extracellular domains [16]. Recently, the determination of the structure of OX2R associated with OxB by electron microscopy has suggested the existence of one key residue (Q134) present in the OX2R orthosteric site, which would be responsible for the activation or inactivation of the receptor [25].

## 3. Orexins and Digestive Cancers

### 3.1. Colon Cancer

Among digestive cancers, colorectal cancer (CRC) represents the third most common cancer world-wide and the third highest cause of cancer-related mortality, which is responsible of about 10% of total cancer death [26]. CRC development resulted from the transformation of normal epithelium to adenoma and then adenocarcinoma. This transformation was associated with multiple genetic and epigenetic alterations consisting of chromosomal instability and microsatellite instability that led to damage in tumor suppressor genes such as *Apc*, *Kras*, *Smad*, *Cdc*, *Tp53*… [27] which dysregulated important intracellular signaling pathways. Moreover, epigenetic alterations (CpG methylation, histones acetylation) also caused a gene dysregulation [27]. If surgery was often the first line of treatment for early-stage cancers, then in more advanced metastatic cancers, chemotherapy based on fluoropyrimidines such as 5-fluorouracil (5-FU), oxaliplatin and irinotecan was proposed [28]. It should be noted that recent treatments based on immunotherapy (anti-PDL-1) and targeted drug therapy can be associated with chemotherapy [29]. Among genetic/epigenetic remodeling in cancer cells, the aberrant expression or the inhibition of various proteins’ expression had a direct impact on cancer cells in terms of proliferation, apoptosis, cell signaling pathways, etc., but also opened the door to the identification of new targets that may lead to new therapeutic approaches. GPCRs represented a class of surface proteins whose expression was modulated in cancer cells by underexpression or overexpression [30]. Moreover, these GPCRs were involved in many important signaling pathways able to play a role in cancer cell proliferation, metabolism and metastasis [31]. In 2004, we demonstrated that OX1R was ectopically expressed in colon cancer and neuroblastoma in which the activation of these receptors by orexins induced an inhibition of cell growth [9]. The percentage of cells from colon tumors expressing OX1R was about 50 to 100% and was independent of tumor location and Duke’s stage. In contrast, OX1R was not expressed in normal colonic mucosa [13]. It should be keep in mind that neither OX2R or orexins were found in colon tumors and normal epithelium. Moreover, OX1R was also expressed in human hepatic metastasis from CRC, indicating that its expression was conserved throughout the epithelial-mesenchymal transition (EMT). OX1R was also expressed in various human colon cancer cell lines, such as HT-29, LoVo, Caco-2, SW620, etc. [13]. It is unknown why OX1R is expressed in colon cancer, although our personal data using methylase/acetylase inhibitors suggest that its expression is dependent on epigenetic regulation (unpublished data). The OX1R activation by OxA or OxB induced a strong inhibition of cell growth (Figure 2) [13]. In the cell growth resulting from a balance between the cell proliferation and apoptosis, our group has demonstrated that orexins did not have any impact on cell proliferation but induced a mitochondrial apoptosis [32]. The deciphering of the mechanism of action by which orexins induced apoptosis identified a new signaling pathway, involving immunoreceptor tyrosine-based motifs (ITIM) and the tyrosine-protein phosphatase non-receptor type 11 (SHP2) (Figure 1). The interaction between orexins and OX1R in colon cancer cells induced the β/γ subunits dissociation from Gq protein, leading to phosphorylation by Src kinases of two ITIM sites present in TM2 and TM7 of the receptor [32,33]. Phosphorylated receptors were able to recruit and activate SHP2, leading to the activation of p38 mitogen-stress protein kinase via RAS/MAPK signaling pathways [34]. These activation cascades induced the translocation of the proapoptotic Bax protein in mitochondria followed by the cytochrome c release involved in apoptosome formation, which led to the activation of caspases 3 and 7, which caused cell death [32,33]. The presence of functional ITIM site in OX1R sequence was not an exceptional situation in GPCR family. Indeed, in bradykinin receptor (B2) and somatostatin receptor (sst2), the presence of ITIM sites associated with SHP2 induced an inhibition of cell proliferation [35,36], whereas in cholecystokinin B receptor (CCK2), this association led to the activation of the AKT signaling pathway [37].

Conventional chemotherapy used in CRC treatment was mainly based on the 5-fluorouracil (5-FU) molecule either associated or not associated with other pyrimidine analogs and/or platinated agents [27]. However, the implementation of chemoresistance mechanisms (primary, before treatment, or secondary, in response to treatment) in CRC and more broadly in digestive cancers including pancreas cancer, hepatocellular cancer, gastric cancer and cholangiocarcinoma, severely limits patient remission. In the HT-29 colon cancer cell line that is resistant to 5-FU (HT-29-FU), OX1R was expressed and orexins induced cell death in these cells, demonstrating that orexin response toward apoptosis was conserved in drug-resistant cancer cells [13]. In preclinical mouse models, subcutaneous injection of colon cancer cells from LoVo or HT-29 cell lines led to the development of tumors. When OxA (or also OxB) was intraperitoneally daily injected, a strong decrease in tumor volume was observed [13]. If the OxA treatment was performed on mice with established xenografted tumors (tumor volume about 150–200 mm^3^), a rapid and strong reversion of tumor volume was identified demonstrating that OxA was able to reduce the established tumors [13]. Histologic analysis of control and OxA-treated tumors indicated that OX1R was similarly expressed along tumor development, showing that the OX1R expression was not modulated by OxA treatment. Moreover, large areas of cell apoptosis, revealed by activated caspase-3 staining, were observed in OxA-treated tumors [13]. In contrast, xenografts obtained with HCT-116 cells which did not express OX1R were totally insensitive to the action of OxA revealed by the absence of tumor volume inhibition [13]. Moreover, the tumor development kinetic was similar between HT-29 cells (OX1R^+/+^) and HCT-116 cells (OX1R^−/−^), indicating that: (i) OX1R expression had no impact on tumor growth in the absence of exogenous orexins; (ii) the presence (or not) of endogenous orexins had no impact on tumor growth; and (iii) the concentration of circulating orexins is too low (about 50 pM) to activate OX1R in tumors [38,39].

### 3.2. Pancreas Cancer

Pancreatic ductal adenocarcinoma (PDAC), which represents over 90% of pancreatic exocrine cancers, is one of the most lethal cancers, with a 5-year survival rate of about 10% [40,41]. Other rare pancreatic exocrine cancers are adenosquamous carcinoma, squamous cell carcinoma and intraductal papillary mucinous neoplasms (IPMN) including colloid carcinoma [42]. According to projections, PDAC could represent the second largest cause of cancer-related deaths in 2030 [43]. The etiology of this cancer was unknown, whereas non-specific risk factors were invoked such as smoking, age, obesity, chronic inflammation, etc. [44]. Genetic alterations characterized by PDAC were *Kras* mutation (over 90% of tumors were *Kras* mutated), *P16/Cdkn2a*, *Tp53*, *Arid1a*, *Brca1/2*, *Smad4*, *hMlh1* and *Msh2* for main mutations [45]. It should be noted that this mutation panel was modulated in other rare pancreas cancer such as IPMN, where the prevalence of *Kras* mutation was found, as well as the prevalence of *Gnas* mutation, which encoded the α_s_-subunit belonging to Gs protein involved in the activation of adenylyl cyclase [45]. The poor prognosis of PDAC was related to the late stage of diagnosis involving systemic metastasis for over 50% of patients. About 20% of PDAC can be surgically resectable [42], frequently associated with neoadjuvant treatment. Unfortunately, the incidence of relapse remained high (over 75%) needing chemotherapy. At this time, first line/second line chemotherapeutic treatment of advanced cancers regrouped two combinatorial regimens differing from country to country and based on Nab/GEM (Nab-paclitaxel/Gemcitabine) or FOLFIRINOX (folinic acid, 5-FU, irinotecan and oxaliplatin). However, these treatments were not well tolerated by patients and the survival gain remained relatively modest [42]. Moreover, chemoresistance resulting from metabolic reprogramming and/or genetic/epigenetic modifications of tumor and/or stromal cells appeared in PDAC [46]. In this context, identification of new targets represents an essential challenge. In 2018, it was reported that OX1R was highly expressed in 96% of 73 tested pancreatic tumors [47]. This expression was not correlated to patient age, tumor stage, tumor size, tumor differentiation and presence or not of metastasis [47]. OX1R was also expressed in precancerous lesions named intraductal papillary mucinous neoplasms (PanIN) with a gradient from low to high, dependent on PanIN grade (PanIN-1 to PanIN-3, respectively). OX1R was not expressed in normal exocrine tissue and OX2R was not expressed in pancreatic normal and tumoral tissues [47]. Moreover, OX1R was expressed in AsPC-1 cell line, which was obtained from nude mouse xenografts initiated with cells from ascites of a 62-year-old patient with PDAC [48]. Activation of OX1R by OxA in AsPC-1 cells induced a drastic inhibition of cell growth resulting in mitochondrial apoptosis, as previously described in colon cancer [13,47]. In preclinical mouse models, OxA reduced the tumor growth in nude mice subcutaneously injected with AsPC-1 cells [47]. Similarly, if isolated cells from a PDAC patient named patient-derived xenograft (PDX) were subcutaneously injected to nude mice, intraperitoneal (ip) injection of OxA also induced an inhibition of tumor growth (Figure 2). Furthermore, OxA treatment started 15–20 days after tumor development of AsPC-1 or PDX cells xenografted in nude mice; a strong and rapid decrease in established tumor volume was observed [47]. As mentioned in the introduction chapter, a lot of OxA antagonists have been developed for the insomnia treatment. Surprisingly, almorexant and also suvorexant were able to inhibit the AsPC-1 cell growth by induction of mitochondrial apoptosis [47]. In the same manner, ip injection of almorexant induced an inhibition of tumor growth in preclinical models (Figure 2). Taking these observations into account, almorexant and suvorexant, which displayed antagonist properties towards the Ca^2+^ signaling pathway, were full agonists able to activate the SHP2-dependant apoptosis signaling pathway in cancer cells [47]. The ability of these ligands to discriminate some signaling pathways were defined as biased ligands [49]. The molecular explanation related to the ability of these antagonists to activate the ITIM/SHP2 signaling pathway in various cancers is currently unknown. However, the recent determination of the structure of orexin receptor X-rays, associated with structure–function relationship studies (see above) should ensure that the role of some OX1R binding site amino acid residues in this activation is understood. It should be noted that activation of the ITIM/SHP2 signaling pathway was dependent on β/γ subunits of Gq protein and independent of αq subunit [4]. 

### 3.3. Gastric Cancer

One group showed that OX1R was expressed in GBC-823 gastric cancer cells line. The activation of OX1R expressed in GBC-823 cells induced an inhibition of apoptosis via the AKT signaling pathway [50]. These observations indicated that the Orexins/OX1R system could have a different behavior related to cancer type (Figure 2). Nevertheless, it is important to note that GBC-823 and SGC-7901 cell lines were problematic because this cell line was contaminated by HeLa cells [51,52], making these observations unreliable.

## 4. Orexins and Other Cancers

### 4.1. Prostate Cancer

Prostate cancer (PC) represents the second most commonly diagnosed cancer among men worldwide [53]. Despite the development of various therapies which are excellent for patients with localized tumors, patients with metastatic advanced prostate cancer have a 5-year survival rate of about 30% [54]. The etiology of PC was elusive but was associated with risk factors including age, ethnicity (higher risk for African-American and Caribbean men), geography (less frequent in Asia, Africa and Central and South America) and family history, and, with less clear effects, diet, smoking, obesity, prostate inflammation and chemical exposure [54]. Genetic alterations found in PC were divided into two groups; inherited gene mutations, encompassing *Brca 1/2*, *Hoxb13*, *Atm*, *Atr*, *Nbs1*, *Chek2*, *Palpb2* and R*ad51d*, and acquired gene alterations, leading to androgen receptor amplification, *Pten* deletions, PI3K/Akt/mTOR pathway alterations, *Tp53* mutations, and *Tmprss2-erg* gene fusions [55,56]. For localized non-metastatic cancers, an active surveillance or local ablation by surgery or radiotherapy were planned [54]. For advanced metastatic cancer, androgens stimulating the tumor development, the first line of treatment consisted of androgen deprivation therapy (ADT), induced by LHRH analogs [57], inhibitor of androgenic steroids synthesis, androgen receptor signaling inhibitors (ASRIs). However, in some patients, a resistance even to low testosterone levels develops, which induces the establishment of castration-resistant prostate cancer (CRPC), which is either associated with metastasis (mCRPC) or not [54]. For CRPC or mCRPC, the treatment with chemotherapeutic agents such as dodetaxel or cabazitaxel, associated with abiraterone or other inhibitors of steroid synthesis, as well as radium-223 in the case of bone metastasis was proposed [58]. More recently, the development of new therapeutic approaches, including a new generation of androgen antagonists that have a greater affinity and no agonist activity for receptors, such as poly ADP-ribose polymerase (PARP) inhibitors, radiopharmaceutical agents as radium-223 or more recently lutetium-177, which allows systemic delivery of radiotherapy and immunotherapy agents as immune checkpoint inhibitors as PD-1 inhibitor offer promising therapies in CRPC [54,59,60]. Among potential targets in the PC treatment, some GPCRs were involved in the progression and development of PC [61]. These receptors associated with their ligands promoted migration, proliferation, neuroendocrine differentiation, mitogenic signaling and invasion of PC [61]. Some small molecules have been tested to block these GPCRs as antagonists of gonadotropin-releasing hormone receptor (GnRH), named degarelix or endothelin A receptor antagonist under clinical trial [54,62]. Inversely, few GPCRs and their ligands were able to inhibit growth of prostate cancer cells. In high-grade advanced cancer (CaP), OX1R, but not OX2R, was highly expressed, but at a lower expression in low-grade prostate cancers [12]. In contrast, OX1R was not expressed in benign prostatic hyperplasia [12]. Preproorexin and OxA expressions were found in “fiber-like” stroma of prostate cancer tissues. OxA was expressed in follicular exocrine epithelium; however, large areas of normal prostate epithelium did not express OxA [63]. It should be noted that OxA was never detected in tumoral tissue, suggesting that OX1R present in tumoral tissue was not activated by endogenous OxA [12]. OX1R was expressed in prostate cancer cell line DU-145 corresponding to androgen-unresponsive cells and also in androgen-responsive cell line LNCaP [12,64]. The activation of OX1R by OxA inhibited the cell growth in these two cell lines [12,64]. In preclinical mouse model obtained by subcutaneous xenografts of DU-145 cells, ip injection of OxA induced a reduction in tumor volume (Figure 2) [65].

### 4.2. Other Cancers

The interaction of OX1R and OX2R with OxA and OxB activated the intracellular Ca^2+^ release through the Gq protein and its αq subunit [6]. Moreover, this interaction also induced the SHP2-dependant mitochondrial apoptosis in cancer cells [33]. However, some groups have identified that the orexins/OXR system was able to activate other signaling pathways (Figure 1), encompassing MAPK-Erk1/2, Pi3K-Akt, adenylyl cyclase/cAMP and JNK [66]. These “alternative” signaling pathways promoted by orexins could also play a role in cancer [67]. Furthermore, orexin’s actions and the expression of the orexin receptor in various cancers were weakly studied, in relation with the poor availability of molecular tools, in particular antibodies directed against receptors or orexins, which were not always specific. Nonetheless, OX1R was expressed in the neuroblastoma cell line, SK-N-MC, and its activation by orexins induced SHP2-dependant apoptosis [9]. However, OX2R was expressed in endometrial carcinomas [68], cortical adenomas [69] and pheochromocytomas [70,71] (Figure 2).

## 5. Conclusions

A link between various cerebral disorders (narcolepsy, Alzheimer’s disease and Parkinson’s disease in which the orexins/OX receptors system was deregulated) and cancers had been suggested [14,71]. The presence of OX1R at the cell surface of various cancers combined to pro-apoptotic actions of orexins in cancer cells could represent a new therapeutic target in the fight against cancer [72]. The development of new molecules including small molecules and/or synthetic antibodies will depict a new challenge for the future decade.

## Figures and Tables

**Figure 1 molecules-26-04849-f001:**
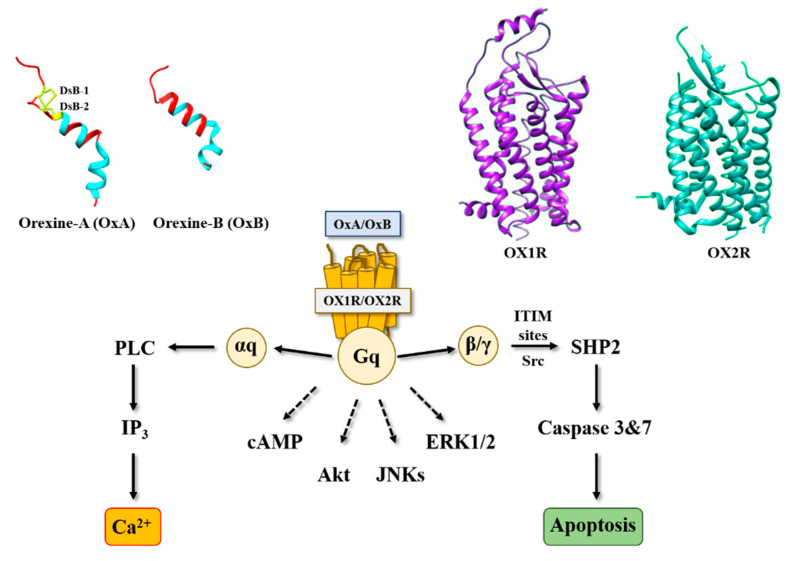
Structure of orexins and their receptors and main signaling pathways activated by orexins in cancer cells. DsB, disulfide bridge; IP3, inositol tri-phosphate; PLC, phospholipase; SHP2, Src homology 2 (SH2) domains of SH2-containing phosphatase 2; cAMP, cyclic adenosine monophosphate; Akt, protein kinase B; JNKs, c-jun N-terminal kinases; ERK1/2, extracellular signal-regulated kinase 1 and 2; ITIM, immunoreceptor tyrosine-based motifs; Src, Src kinases.

**Figure 2 molecules-26-04849-f002:**
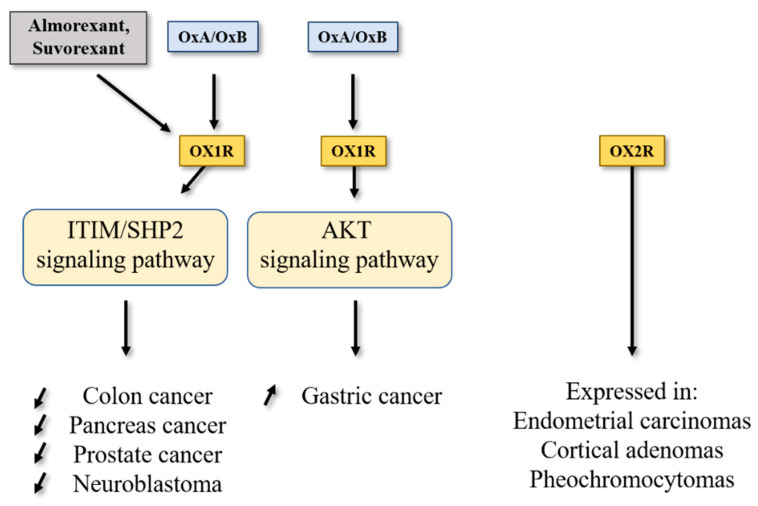
Impact of orexins/OXR system on cancers. SHP2, Src homology 2 (SH2) domains of SH2-containing phosphatase 2; Akt, protein kinase B; ITIM, immunoreceptor tyrosine-based motifs.

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
