# Peer review of "Orexins/Hypocretins and Cancer: A Neuropeptide as Emerging Target"

_molecules, 2021, doi:10.3390/molecules26164849_

Round 1

Reviewer 1 Report

Manuscript: Orexins/Hypocretins and cancer: A neuropeptide as emerging target.

Authors: Alian C et al.,

The Orexin system has diverse functions in a normal and healthy life. The absence or excess amount of orexin causes many neurological disorders. The authors focusing on the role of orexins/OX1R in different types of cancers.

The article is more focused on the role of orexin and orexin receptor 1 in colon cancer, pancreatic cancer gastric cancer. Also, the authors explain the role in prostate and other types of cancers.

This article gives the awareness of the orexin system for new therapeutic challenges and treatment of the different types of cancers.

The article is well written and organized. 

Comments:

A graphical summary will be more attractive for readers.

Reference number not mentioned line 37.

Reference order not proper in line 57.

Author Response

Replies to Reviewer 1:

We warmly thank the reviewer for taking the time to review our manuscript and for offering several insightful suggestions.

All modifications were highlighted in yellow in the revised manuscript.

A graphical summary will be more attractive for readers.

A new figure was added to this revised manuscript, to summarize the orexin impact on cancers (see figure 2). Reference to this new figure 2 was added to Page 7, line 139; Page 10, line 216; Page 11, line 237 and Page 13, line 283 and line 297.

Reference number not mentioned, line 37.

The reference number was now added (ref. 6)

Reference order not proper in line 57

The sentence corresponding to the cited reference (line 53-54 assuming new numbering) was modified as:

“Since the 2000s, it was shown, in pathological conditions, that OX1R (but not OX2R) was abnormally expressed in peripheral cancers (9).”

Reviewer 2 Report

This review covers a lot of information on hypocretin/orexin receptors as potential targets in the treatment of various cancers. There is a lot of good info in the manuscript, but the writing needs to be improved substantially as it is quite hard to understand in its present form. Additional critiques are listed below:

  • Pg. 2 ln 65: Should mention that the orexins are encoded by the same gene (hcrt) but protein isoforms are generated through alternatively spliced mRNA species.
  • There is a lot of jumping around between genetics, epigenetics, signaling pathways, and structural biology, with little connections between these areas. A reorganization of the manuscript walking through how these facets of cancer/orexin signaling are related would substantially improve the flow of the writing. 
  • Pg. 3 ln 117: these gene names should be lowercase/italicized 
  • A general discussion/speculation as to why (proximate/ultimate explanations) various cancers ectopically express OxRs would be beneficial to put the material in context.
  • It is still unclear why receptor antagonists (almorexant/suvorexant) and receptor agonists (orexins) would both elicit apoptosis in cancer cells. 
  • Is there any cancer cell-extrinsic mechanisms involved in the effects of OxR signaling? Do immune or other stromal cells participate in the effects of OxR agonism/antagonism on tumors? Is there any info available on this? This is especially relevant given the ITIM/SHP2 interactions discussed, ideally this would influence immune cells.
  • It is unclear why the authors focus on a relationship (e.g., in the conclusion) between cerebral disorders (narcolepsy, Alzheimer's disease...) and peripheral OxR signaling in cancer....there is no evidence to suggest that these are related as far as I know.
  •  

Author Response

Replies to reviewer 2:

We warmly thank the reviewer for taking the time to review our manuscript and for offering several insightful suggestions.

All modifications were highlighted in yellow in the revised manuscript.

  • 2 ln 65: Should mention that the orexins are encoded by the same gene (hcrt) but protein isoforms are generated through alternatively spliced mRNA species.

The corresponding sentence (lines 34-35 assuming new numbering of revised manuscript)) was modified according rewiever’s remark.

  • There is a lot of jumping around between genetics, epigenetics, signaling pathways, and structural biology, with little connections between these areas. A reorganization of the manuscript walking through how these facets of cancer/orexin signaling are related would substantially improve the flow of the writing.

The manuscript was organized in various chapters including a chapter dedicated to the very recent data on structure of orexins and their receptors. We believe that these aspects could be interesting for the reader. Moreover, as suggested by the reviewer in their different remarks, we have included new sentences about the surprising pro-apoptotic role of antagonists (see below). These new sentences refer to this chapter.

Genetics and epigenetics aspects refer only to the description of various cancers described in this review in term of ethiology, risk factors, treatments…. But never refer to orexin action. We believe that is important to describe each cancer where orexins have an anti-tumoral action before to describe this action.

  • 3 ln 117: these gene names should be lowercase/italicized 

Sentences having gene names have been modified according rewiever’s remark.

  • A general discussion/speculation as to why (proximate/ultimate explanations) various cancers ectopically express OxRs would be beneficial to put the material in context.

Unfortunately, the cause of ectopic OxRs expression in cancers was unknown. However, our unpublished data suggest that this expression is regulated by epigenetic modifications. Indeed, the use of methylase/acetylase inhibitors are able to modulate this ectopic expression. In this respect, we have added, in this review, one sentence (page 7 lines 136-138) suggesting this aspect.

  • It is still unclear why receptor antagonists (almorexant/suvorexant) and receptor agonists (orexins) would both elicit apoptosis in cancer cells. 

We agree with the reviewer, but we have published that almorexant and suvorexant are able to induce apoptosis in pancreas cancer. Moreover, we observed the same properties with other antagonists as lemborexant (unpublished data). We believe that the structure-function relationship studies of OX1R will allow to understand this aspect.

We have added three sentences (pages 10-11, lines 226-232) related to this aspect in revised review manuscript.

  • Is there any cancer cell-extrinsic mechanisms involved in the effects of OxR signaling? Do immune or other stromal cells participate in the effects of OxR agonism/antagonism on tumors? Is there any info available on this? This is especially relevant given the ITIM/SHP2 interactions discussed, ideally this would influence immune cells.

Unfortunately, no information was available about the role (or not) of orexins on immune and/or stromal cells in cancers. In contrast, the anti-inflammatory role of orexins was mediated by immune cells but these actions were mainly mediated by the Ca2+ release signaling pathway independently of ITIM/SHP2 interactions.

  • It is unclear why the authors focus on a relationship (e.g., in the conclusion) between cerebral disorders (narcolepsy, Alzheimer's disease...) and peripheral OxR signaling in cancer....there is no evidence to suggest that these are related as far as I know.

This aspect refers to very recent publications from Ferri’s group (Sleep Med Rev. 2021,56,101409 and Cancers (Basel) 2021,13,2612.) whose established a relation between neurodegenerative diseases, orexins and cancers.

Some extracts of these reviews:

The studies on the potential modulation of the orexin system in cancer and in neurodegenerative diseases are still pioneering and further human data are needed, although they have already shown promising results.

Taken together, these different lines of evidence have a relevant possible translational value, which might lead to the arrangement of novel therapeutic approaches to both neurodegenerative disease and cancer by modulating orexin pathways. This perspective warrants an important research effort on this topic in the near future.

We believe that these aspects could interest readers and represent an interesting perspective.

Round 2

Reviewer 2 Report

The authors have addressed my prior concerns, although I recommend additional editing for clarity. 

Author Response

Replies to reviewer 2:

We thank again the reviewer for taking the time to review our manuscript.

All modifications were highlighted in yellow in the revised manuscript.

  • The authors have addressed my prior concerns, although I recommend additional editing for clarity.

To clarify and assist the reader, we have modified the sentence (page 4, lines 62-64) to define the outline of this review.

In addition, after proofreading by an English-speaking citizen, we have corrected some errors and typos (highlighted in yellow).